# Caregiver Perceptions of Environmental Facilitators and Barriers to Healthy Eating and Active Living during the Summer: Results from the Project SWEAT Sub-Study

**DOI:** 10.3390/ijerph182111396

**Published:** 2021-10-29

**Authors:** Laura C. Hopkins, Amy R. Sharn, Daniel Remley, Heather Schier, Regan Olak, Dorsena Drakeford, Cara Pannell, Carolyn Gunther

**Affiliations:** 1Department of Public Health and Prevention Sciences, College of Education and Health Sciences, Baldwin Wallace University, 328D Malicky Center, 275 Eastland Road, Berea, OH 44017, USA; lhopkins@bw.edu (L.C.H.); rolak19@bw.edu (R.O.); ddrakefo20@bw.edu (D.D.); 2Department of Human Sciences, The Ohio State University, Human Nutrition Program, Campbell Hall, 1787 Neil Avenue, Columbus, OH 43210, USA; sharn.3@osu.edu (A.R.S.); schier.8@osu.edu (H.S.); pannell.59@osu.edu (C.P.); 3Family and Consumer Sciences, The Ohio State University, OSU Extension, 1864 Shyville Road, Piketon, OH 45661, USA; remley.4@osu.edu

**Keywords:** summer, food environment, physical activity environment, qualitative

## Abstract

Objective: The aim of this study was to examine caregiver perceptions of summertime neighborhood-level environmental barriers and facilitators to healthy eating and active living in their elementary-age racial minority children. Methods: Caregivers with students in the prekindergarten–fifth grade were recruited from two schools located in low-income urban neighborhoods of Columbus, OH, with a predominantly Black population. Participants engaged in the research portion of the Healthy Eating Active Living: Mapping Attribute using Participatory Photographic Surveys (HEALth MAPPS^TM^) protocol, which included (1) orientation; (2) photographing and geotagging facilitators and barriers to HEALth on daily routes; (3) in-depth interview (IDI) discussing images and routes taken; (4) focus groups (FG). IDIs and FGs were transcribed verbatim. Analyses were guided by grounded theory and interpretive phenomenology and were coded by researchers (*n* = 3), who used comparative analysis to develop a codebook and determine major themes. Results: A total of 10 caregivers enrolled and 9 completed the IDIs. Five caregivers participated in focus groups. A majority (77.8%, *n* = 7) of caregivers identified as Black, female (88.9%, *n* = 8), and low income (55.6%, *n* = 5). IDI and FG themes included (1) walkway infrastructure crucial for healthy eating and active living; (2) scarce accessibility to healthy, affordable foods; (3) multiple abandoned properties; (4) unsafe activity near common neighborhood routes. Conclusions: Caregivers perceived multiple neighborhood-level barriers to healthy eating and activity during the summer months when school is closed. Findings from this study provide initial insights into environmental determinants of unhealthy summer weight gain in a sample of predominantly racial minority school-age children from low-income households.

## 1. Introduction

The summertime and other periods of time when school is closed or out of session is a period of risk for accelerated weight gain among elementary school-aged children [1,2,3,4,5]. Children who are already overweight or obese and those who identify as non-Hispanic Black or Hispanic are at greatest risk for this health threat risk for unhealthy summer weight gain [1,2,3,4,5,6]. Some studies suggest that nearly all of the increase in body mass index (BMI) from one year to the next occurs over the summer [1,6,7] and that children are most likely to transition into being overweight or obese during the summers after kindergarten and second grade [8]. Our preliminary work [9] and others [2] suggest that BMI decreases when children return to school after the summer break but not to baseline levels. Therefore, summer weight gain could contribute to long-term childhood overweight and obesity.

It is well established that a child’s risk for obesity is influenced by their food (e.g., school food environment, neighborhood food access, food present in the home, etc.) and physical activity (e.g., availability of physical activity equipment, access to recreational areas (e.g., parks, recreation centers, etc.) environments [10,11,12,13,14,15,16]. Thus, it is plausible from a theoretical standpoint that inappropriate summertime weight gain is due, at least in part, to a shift in the food and physical activity environment to which children are exposed, whereby they transition from spending their waking hours in the school setting during the academic year (access to healthy meals and snacks via USDA’s child meal programs; access to safe play and structured time spent in physical activity) to the home/neighborhood during the summer when they lose access to the school’s physical resources. Unfortunately, there is a paucity of research on the extent to which these environmental factors contribute to the problem. Without this information, it will be difficult to evaluate the effectiveness of and inform enhancements to existing policy and programs (e.g., USA: Summer Food Service Progam, National School Lunch and Breakfast programs) aimed at addressing the problem. 

Project Summer Weight and Environmental Assessment Trial (SWEAT) (ClinicalTrials.gov Identifier: NCT03010644) was an observational, prospective study exploring weight and health trends during the summer months among elementary-aged racial minority children residing in low-income, urban neighborhoods [17]. The objective of this sub-study of Project SWEAT was to examine the summertime facilitators and barriers to healthy eating and active living. 

## 2. Materials and Methods

### 2.1. Study Design

Project SWEAT was implemented at 2 elementary schools in predominantly Black low-income urban neighborhoods of Columbus, OH [17]. The study encompassed (1) a main study exploring child weight and health trends and (2) a sub-study employing mixed methods approaches examining the food, physical activity, and social-behavioral, and environmental determinants of unhealthy weight gain during the summer months. The data presented in this article are from the Project SWEAT sub-study, specifically, the sub-study focused on the neighborhood-level food and physical activity environment. Sub-study data collection timepoints included baseline (T0, beginning of summer), time point 1 (T1; midsummer), and time point 2 (T2; the beginning of school year). The sub-study occurred from June to September 2017. 

### 2.2. Participants and Recruitment

All children and their primary caregivers in prekindergarten (pre-K) through fifth grades enrolled at the two participating schools were invited to participate in the Project.

SWEAT main study at the end of school year 1. At the time of the study, the schools combined had 794 students enrolled (*n* = 460 and *n* = 334). Indirect and direct recruitment methods were employed. For indirect recruitment, an informational sheet describing the study and a demographic survey was sent home with each child in pre-K through fifth grades. For direct recruitment, study staff attended school events and were present at child drop-off and pickup times to speak with caregivers about the study directly. Return of a completed demographic survey indicated permission from caregivers to enroll their child or children in the main study. All participants who enrolled in the main study were contacted via telephone and invited to participate in the sub-study. All families who enrolled in the sub-study (*n* = 62 children representing 39 families) were invited via in-person and text message to participate in the HEALth MAPPS^TM^ sub-study. Thus, the sample of participants was a convenience sample. 

### 2.3. Data Collection

Data collectors were undergraduate and graduate students from nutrition, dietetics, or other related fields. All underwent a two-hour didactic training session. Data collectors were then observed by a senior researcher until proficiency in methodology was achieved. 

All data collection occurred at participant homes or community locations (e.g., library, school, recreation center). Caregiver consent, caregiver permission, and child assent for the substudy during Phase 1: Orientation (Figure 1) was obtained. Each dyad (child[ren] and caregiver) received 20 dollars in cash to purchase food while on their routes.

All study materials and procedures were approved by The Ohio State University Behavioral and Social Sciences Institutional Review Board (2016B0034).

### 2.4. Outcome Measures

#### 2.4.1. Household Demographics

On the Project SWEAT information sheet, caregivers were asked to complete a brief demographic survey with questions pertaining to (1) the caregiver’s age, sex, race, and ethnicity^1^, (2) annual household income, and (3) household food security [19]. 

#### 2.4.2. Neighborhood-Level Food and Physical Activity Environment

The neighborhood-level food and physical activity environments were assessed qualitatively by examining facilitators and barriers to healthy eating and physical activity patterns through the caregiver and child perspective. A portion of the Healthy Eating and Active Living: Mapping Attributes Using Participatory Photographic Surveys (HEALth MAPPS^TM^) protocol^8^ was adapted to the current study and included photography, in-depth interviews, and focus groups in 4 distinct phases (Figure 1). 

### 2.5. Data Analysis

Demographic information was coded as follows: For race, participants were classified as Black or non-Black. Participants were classified as Black if they reported being African American or African American and another race. All others were classified as non-Black, which included non-Hispanic White and Hispanic White. For household income, a binomial variable was created. Annual household income data were collected categorically: (a) USD < 10,000; (b) USD 10,001–20,000; (c) USD 20,001–30,000; (d) USD 30,001–40,000; (e) USD 40,001–50,000; (f) USD 50,001–60,000; (g) USD 60,001–80,000; (h) >USD 80,000. Based on responses to the annual household income question, participants were assigned an income level based on the mid-point between the income range. For example, if a participant responded that their annual household income was between USD 10,001 and USD 20,000, they were assigned an income level of USD 15,000. This annual household income level was compared with the national poverty guidelines [20] and based on the number of individuals living in the household, participants were classified as low-income or non-low-income. For household food security, raw scores were calculated and categorized as marginal or high food security, low food security, or very low food security, according to the USDA’s US Household Food Security Survey Module: Six-Item Short Form [19].

Audio recordings of in-depth interviews and focus groups were transcribed verbatim. Data analysis for in-depth interviews and focus groups were guided by grounded theory and interpretive phenomenology [21,22]. Researchers (*n* = 3) independently conducted line-by-line open coding for all in-depth interviews and focus groups. Through the process of constant comparative analyses, a codebook was developed, and themes were identified for the in-depth interviews and focus groups [21,22]. 

## 3. Results

### 3.1. Participants

In total, 10 families enrolled in the HEALth MAPPS^TM^ sub-study. A consort diagram of participant completion of HEALth MAPPS^TM^ phases is provided in Figure 2. The HEALth MAPPS^TM^ caregiver sample was not significantly different from the Project SWEAT main study sample (Table 1). 

### 3.2. HEALth MAPPS^TM^ Themes

Eight themes and three subthemes arose from the analyses (Table 2).

*Community Resources as HEALth Facilitators*: Within the neighborhood, participants mentioned in IDIs and FGs that food support available at local schools and community centers improved access to healthy foods during the summer months. Participants also cited the local community center that provided cooking lessons, programming, and nutrition education allowed them and their neighbors to live healthier lives.*Personal Motivations for Improving the Community and their Lifestyle are HEALth Facilitators:* From the photos taken on their routes, caregivers’ IDIs described that taking ownership in the community was a HEALth facilitator. Caregivers shared that they wanted to improve areas that were abandoned and overgrown with vegetation to replace them with parks and other community resources and were also aware of the gentrification happening in their neighborhoods. In addition to wanting to improve their community as part of their personal motivations, they also wanted to improve their grocery shopping habits, often citing that the nutritionally poor diet they grew up on was not something they wanted to continue to pass down to their children.*Availability and Access (or lack of) to Safe Physical Activity are HEALth Facilitators/Barriers:* When playgrounds and parks were in the area, caregivers shared in their IDIs that they felt it was much easier for their children to have safe physical activity, given that they were well maintained. Outside of the physical activity resources within their neighborhoods, caregivers cited that the cost to belong to a gym, high age admittance to summer camps, and limited time made it difficult to provide for themselves and their children for safe physical activity.*Lack of Availability of Healthy Food is a HEALth Barrier:* Several caregivers in IDIs cited that the foods available to them in their neighborhoods, local fast food establishments, and grocery stores made it difficult for them to eat healthfully. Caregivers also cited a plethora of “junk food” that was available in their community either in grocery or corner stores and that the food offered to them at food banks or grocery stores was often rotten or expired.*Food Access (Cost) as a HEALth Barrier:* Another common barrier to HEALth was food cost, mentioned in both IDIs and FGs. Caregivers cited the difficulty in balancing the costs of daily living with increased grocery budgets to include foods that are both nutrient and calorically dense. They felt that this difficulty was exacerbated during the summer months when children are not receiving meals at school. Inconsistent pricing among stores, increased costs of healthy foods, and budget restraints were the common barriers mentioned that inhibited participants from purchasing healthy foods. Participants noted that the increased costs of healthy foods were present among the stores they frequently shopped at, as well as the internet, and that having to shop around for the best prices made it difficult to stay within their budgeted dollar and time amounts for food shopping.*Time Constraints as a HEALth Barrier:* Another common barrier to HEALth was time constraints, mentioned in both IDIs and FGs. Caregivers cited that in order for them to prepare, serve, and eat balanced meals, they felt that had to give up time toward active living and household errands. Having to drive across town to be able to purchase healthier foods that were not readily available in their neighborhood was also a time constraint in healthy eating in their homes. Within FGs, caregivers mentioned that the timing of the year made it difficult to provide healthy food options for their children, as there were more meals to provide for. Conflicting schedules with children’s sports and family mealtime made it difficult to procure and prepare healthy food during the summer months. Additionally, caregivers cited that in order to prepare a healthy meal, they felt they needed to give up time dedicated toward physical activity and vice versa. One facilitator to preparing a healthy meal was the use of a crockpot, mentioned in the focus group, which helped reduce the time spent in the kitchen.*Nutrition Knowledge (or lack of) as a Facilitator/Barrier to HEALth*: Caregivers also communicated in IDIs and FGs that their limited nutrition and food safety knowledge made it difficult for them to prepare healthy meals and select unspoiled produce. The local community center that provided parenting and cooking classes was a facilitator to HEALth and helped them to increase their nutrition knowledge in preparing healthy family meals. This presence of strong community programming helped in creating self-efficacy for caregivers in the neighborhoods to obtain and prepare healthy meals in their homes.*Neighborhood Safety as a Barrier to HEALth:* Among caregivers, the most mentioned barrier to HEALth was neighborhood safety among IDIs and FGs. Caregivers described the presence of multiple abandoned lots as homes that made themselves and their neighbors uneasy about the safety of walking through their neighborhoods for HEALth. Trash among their neighborhoods on abandoned lots with glass, drug and alcohol paraphernalia, and dead rodents made them uncomfortable about accessing HEALth for themselves and their children.

A subtheme within the neighborhood safety theme within IDIs and FGs was *Poorly Maintained or Absence of Infrastructure as a Barrier to HEALth:* Multiple caregivers cited that poorly maintained or absence of sidewalks made it difficult for themselves and their neighbors to access both healthy food and safe physical activity. When adequately maintained sidewalks were present, caregivers stated that it was easier for both physical activity and access to healthy foods. Participants also mentioned that the presence of long-standing construction projects, low electrical wires, and lack of well-maintained sidewalks in their neighborhoods made it difficult for their children to participate in safe physical activity. For physical activity and healthy eating, caregivers cited that playgrounds, sidewalks, and bike lanes were not well maintained or designed for their children to navigate their neighborhood safely with traffic. These dangers were cited as reasons that caregivers felt that they could not allow their children to safely navigate the neighborhood.Perception of crime and low safety in neighborhoods inhibited HEALth, presenting the second subtheme within neighborhood safety from the FGs, with *Crime as a Barrier to HEALth*. Caregivers mentioned that high instances of traffic, abandoned homes, vandalism, and criminals living in the area made caregivers feel uneasy about their children leaving the home for physical activity. These concerns led to perceived decreased amounts of physical activity in their neighborhoods.Due to these safety concerns, caregivers within IDIs shared that the *Need for Adult Supervision is a Barrier to HEALth*. Caregivers wanted to be able to walk their children to and from the community center where they could participate in HEALth, though they did not always have the time to do so. Having the community center supervise the children while they were unable to watch their children was a good resource to provide caregivers with a needed break.

## 4. Discussion

The summertime and other times when school is out of session are windows of risk for unhealthy weight gain and overall health decline among elementary school-aged children. It is well established that food and physical activity environments to which children are exposed contribute to childhood obesity. Thus, it is likely that summertime environments may contribute to unhealthy weight gain and overall health decline that has been observed among elementary school-aged children. However, research is limited regarding child environments during the summer months that may be contributing to accelerated summertime weight gain. With the US Congress slated to review the Child Nutrition Reauthorization Act in early 2022, the results from the current study are ever more salient to informing policymakers and researchers alike on the facilitators and barriers to healthy eating and physical activity during times when school is out of session. 

Food deserts, characterized as areas with limited access to retail food stores and often among low-income communities, pose unique barriers to healthy eating and may increase the risk for child malnutrition [23]. Within the literature, Black caregivers of children 3–11 years of age have noted that their food environments are marketed toward supporting unhealthy eating practices, and this may create additional barriers in creating healthy eating environments for Black youth [24], especially given the autonomy that adolescents gain in food purchasing and eating behaviors. Though food may be more readily available for unhealthy eating, caregiver perceptions of the cost of fruits and vegetables may also hinder caregiver and children’s ability to make healthy food choices and provide a healthy food environment in their home [25]. The present study revealed that caregivers share similar perspectives as noted in the literature, including barriers to food access (cost and distance to food retail stores), time constraints, and breadth of nutrition and food knowledge. Despite these challenges, Black caregivers are resourceful in procuring foods for their family through available resources (e.g., social support, local and federal food programs) [26]. In concordance, the current study found caregivers demonstrate resourceful problem solving (the literature often refers to these behaviors as “coping mechanisms”. The authors chose to use “resourceful problem solving” to better and more appropriately credit the participants of this study). Examples include employing cooking methods to manage time and promote physical activity (e.g., preparing meals in a crockpot), participating in nutrition and food education (e.g., local cooking classes) and federal food programs (e.g., US National School Lunch Program) to promote perceived healthy eating for children. As cited by a participant, child nutrition policies (e.g., US NSLP), contributes toward household food security. This is corroborated by extensive research [27]. In the US, food programs and services offered to children in the summer are underutilized, rendering the summertime a period of increased risk for food insecurity for families [28].

It has been well established that low-income and predominantly minority neighborhoods have less access to physical activity resources (e.g., parks, recreation facilities, etc.) [29,30,31] and neighborhood spaces tend to be less safe, less well tended, crime ridden, etc. [32,33,34,35,36] While, the literature already establishes the association between low socioeconomic status and prevalence of crime, as well as low socioeconomic status and food accessibility, not all crimes are reported. This presents researchers with challenges when attempting to capture the association between neighborhood crime and food accessibility. In the present study, caregivers were asked to identify barriers to their children’s physical activity. Caregivers recognized the importance of incorporating physical activity in their children’s lives but specifically noted neighborhood safety as a barrier to physical activity. This finding is consistent with other studies in the literature regarding child and adult perceptions of barriers and facilitators to physical activity in low-income neighborhoods. Finkelstein and Peterson sought to understand why children residing in low-income neighborhoods of Colorado experienced lower rates of physical activity, compared with other states [37]. They identified several barriers to participation in physical activity including traffic, illicit activity, and other neighborhood inequities, that prevent their children from being able to engage in outdoor physical activity. This finding has been demonstrated in other peer-reviewed literature as well [38]. The lack of actual and perceived access to safe physical activity environments within a family’s neighborhood presents several challenges for children and their caregivers to engage in adequate levels of physical activity. In the preschool-aged population, the prevalence of a BMI ≥ 95 percentile has been shown to be statistically impacted by the rate of 911 calls made in the child’s neighborhood [39]. Given that during the summer months, children likely lose access to other safe physical activity environments and opportunities (e.g., physical education, school gym facilities, etc.), it is plausible that neighborhood safety may be a contributing factor to the accelerated weight gain of children during the summer months. 

The importance of community resources should not be underestimated in making healthy choices for eating and active living. When the environment is not safe, or food education is not available, families heavily rely on public and community resources to fill the gaps. The data from the current study found that caregivers used resources such as neighborhood recreation centers, to keep children active provide nutrition education and activities, as well as community gardens to explore healthier food options. In previous studies, the use of community-based programs has shown a decrease in chronic illnesses in those who are impacted by poverty, unsafe conditions, or limited access to healthy foods [40]. Thus, during the summertime, when children are at risk for declines in health status, as well as other risks such as food insecurity, the summer academic slide, etc., community resources should be maximized by education, public health, and healthcare professionals to ameliorate these risks.

This study is the first known application of the research portion of HEALth MAPPS^TM^ protocol [18,41] engaging caregivers and children exploring the summertime window of risk in an urban environment. Despite the novelty of this study, limitations did exist, including generalizability, i.e., findings from this study may not be applicable to all urban environments, or non-urban (e.g., rural) environments. Additionally, given the convenience sample of participants from the Project SWEAT main study, which can result in a biased sample. However, as demonstrated in Table 1, there were no differences in baseline participant characteristics between the main study and sub-study samples. Participants also reported limited time to take photos due to a small number of devices of the high cost that were shared among participants. Future studies should investigate the validation of smartphone-based apps that would be able to provide multiple opportunities for participants to MAPP their neighborhoods without time restrictions. Finally, the participants of this sub-study were caregivers who reported their perceptions of barriers and facilitators to healthy eating and active living for their children. Future work should include child participants and a comparison of child and caregiver perceptions. 

## 5. Conclusions

Adapting a portion of the HEALth MAPPS^TM^ methodology for Project SWEAT provided valuable visual insights into the facilitators and barriers to healthy eating and active living for children and adolescents face in underserved, urban communities during the summer months. Upon learning of these visual and geographic determinants, changes can be made to protect against the summer window of risk for unhealthy weight gain [8,42]. Both sites indicated barriers and facilitators to healthy eating and active living. The findings from the images and their perceptions indicate a need for intervention among urban school neighborhoods, and interventions should be tailored to reflect communities’ needs.

This information should be used by local- and state-level stakeholders to improve low-income neighborhood environments in Columbus, Ohio, to promote healthy eating and active living during the summer months. Future efforts should be directed toward learning more about food and physical activity environments across rural and suburban sites and other minorities disproportionately affected by childhood obesity [1,2,3,4,5] to improve the food and physical activity environments during the summer months for children and communities everywhere.

## Figures and Tables

**Figure 1 ijerph-18-11396-f001:**
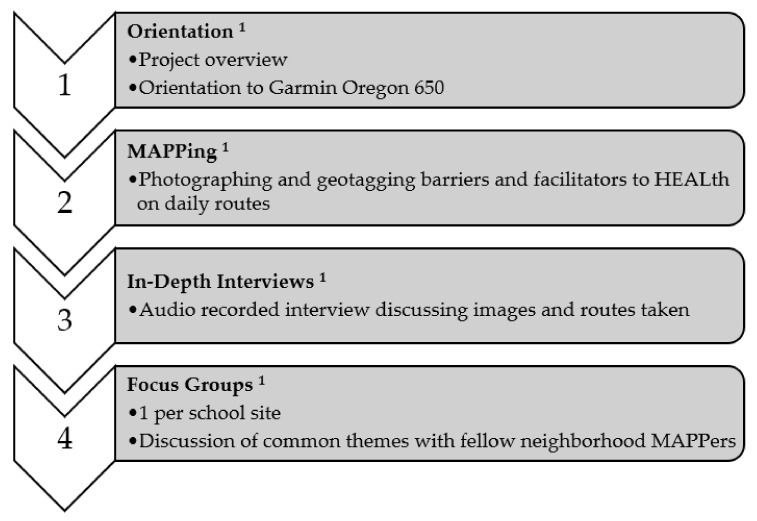
Phases of the Project SWEAT HEALth MAPPS^TM^ Sub-Study: ^1^
*Orientation:* Families were introduced to the HEALth MAPPS^TM^ and Garmin Oregon 650 via an orientation handout to explain the objectives of the project, the participants’ role within the project, and functions of the tracking device. Participants were compensated with USD 20 in cash to purchase food while on their routes. *MAPPing:* Caregivers and their children used the Garmin Oregon 650 together to map as many commonly traveled routes as desired. Participants took images of their facilitators and barriers to healthy eating and active living. Participants were also asked to photograph any food purchases they made while on their routes. *In-Depth Interviews (IDIs):* Caregivers participated in audio recorded in-depth interviews with questions based on the HEALth MAPPS^TM^ protocol [18]. Participants were verbally asked to identify facilitators and barriers to healthy eating and active living within their images taken on their route with the HEALth MAPPS^TM^ Route Journal [18]. *Focus Groups:* Focus Groups were conducted at each respective school site, with a separate focus group for caregivers and children. Caregivers were asked to discuss a series of questions or statements regarding their neighborhoods with researchers (*n* = 2) acting as a moderator and a field note taker.

**Figure 2 ijerph-18-11396-f002:**
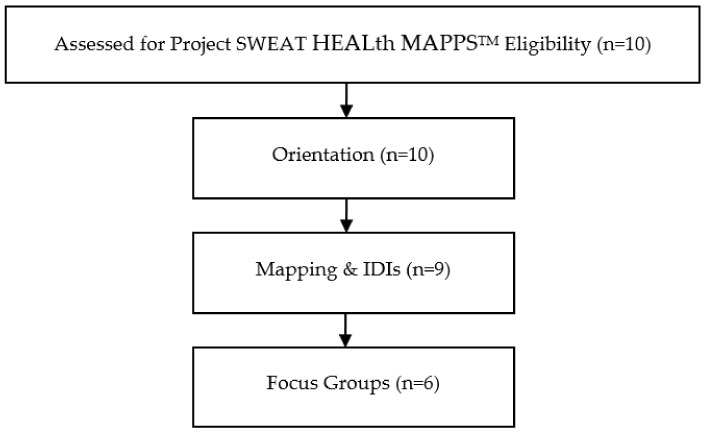
Project SWEAT HEALth MAPPS^TM^ CONSORT flow diagram.

**Table 1 ijerph-18-11396-t001:** Project SWEAT HEALth MAPPS^TM^ sub-study caregiver demographic characteristics.

	Project SWEAT Main Study	HEALth MAPPS^TM^ Sub-Study Sample	*P*
Caregiver Age (years), Mean ± SE	34.69 ± 1.17	38.38 ± 2.43	0.28 ^d^
Caregiver Sex, % (*n*) Female Male	83.33 (65) 15.38 (12)	77.78 (7) 22.22 (2)	0.79 ^e^
Caregiver Race ^a^, % (*n*) Black Non-Black	78.21 (61) 21.79 (17)	77.78 (7) 22.22 (2)	0.97 ^e^
Caregiver Ethnicity. % (*n*) Non-Hispanic or Latino Hispanic or Latino	98.63 (72) 1.37 (1)	100.00 (9) 0.00 (0)	0.71 ^e^
Caregiver Low-Income ^b^, % (*n*) Low-Income Non-Low-Income	67.11 (51) 32.89 (25)	55.56 (5) 44.44 (4)	0.43 ^e^
Caregiver Household Food Security Category ^c^,% (*n*) Very Low Food Security Low Food Security Marginal or High Food Security	14.10 (11) 12.82 (10) 73.08 (57)	11.11 (1) 11.11 (1) 77.78 (7)	0.94 ^e^

^a^ Race = Black or non-Black; Black if the caregiver reported that they were Black or both Black and another race or ethnicity; ^b^ A binomial variable was created. Annual household income data were collected categorically: (a) USD < 10,000; (b) USD 10,001–20,000; (c) USD 20,001–30,000; (d) USD 30,001–40,000; (e) USD 40,001–50,000; (f) USD 50,001–60,000; (g) USD 60,001–80,000; (h) >USD 80,000. Based on responses to the annual household income question, participants were assigned an income level based on the mid-point between the income range. This annual household income level was compared with the national poverty guidelines [20] and based on the number of individuals living in the household, participants were classified as low-income or non-low-income. ^c^ Raw scores for Household Food Security were calculated and categorized according to the USDA’s US Household Food Security Survey Module: Six-Item Short Form [19]. ^d^ T-test. ^e^ Chi^2.^

**Table 2 ijerph-18-11396-t002:** HEALth MAPPS^TM^ Themes.

Representative Quote(s) from Project SWEAT HEALth MAPPS Participant In-depth Interviews and Focus Groups and Image(s) Captured by Project SWEAT HEALth MAPPS Participants using the Garmin Oregon 650
Theme 1. Community Resources as HEALth Facilitators
“So, that affects them physically, and as far as eating, you know they’re not getting the, maybe the whole MyPlate uh proportion of their meal, especially if the parent is not focused on, or not educated enough because that facility actually does all that. Educating parents on how to prepare healthy meals and things of that nature but, if you don’t have a safe place to come do that, then that can affect them in all areas.”	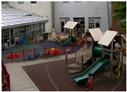
43-year-old, non-Hispanic Black Female
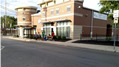	“…A lot of them from mid-Ohio, I’ve noticed a lot of the bigger community centers are using that Mid Ohio Foodbank that comes and delivers a lot of that produce. But there is a lot of, in this area, places that do that. Even the school do that, promoting healthy eating…”45-year-old, non-Hispanic White Female
Theme 2. Personal Motivations for Improving the Community and their Lifestyle are HEALth Facilitators
“…But this is another unwanted area that I see is not being used for nothing, that it would be nice to have, you know, I don’t understand why we, it’s just a lot of vacant areas…That could be used for um either like a garden area, or um something um a little park area for the children…”43-year-old, non-Hispanic Black Female	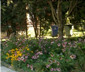
Theme 3. Availability and Access (or lack of) to Safe Physical Activity are HEALth Facilitators/Barriers
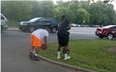	“That’s a little park over there in the corner, it’s not even a big park, it’s just an area they built to set down. The people can walk by, people can sit in there, people can get a pizza by it. It’s really small area.” 36-year-old, non-Hispanic Black Female
“There’s a balance somewhat because there’s activity at school and in summer. But in the summertime there’s a lot more because there’s active summer camps and swimming and there’s a lot of stuff that in the wintertime it dies down and it just… They have to be interested in something that’s offered in the wintertime or encouraged to participate in something at the rec center and not all families have the funds or availability to get to these different things.”	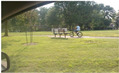
38-year-old, non-Hispanic Black Female
Theme 4. Lack of Availability of Healthy Food is a HEALth Barrier
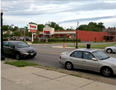	“Either that or they’re going to shop at these little convenient markets, where you run across other things, other kinds of foods. Um but I like the fact that you do have the Save-A-Lot in walking distance in the uh community here, in the [zipcode].” 43-year-old, non-Hispanic Black Female
“It’s harder cause there’s no uh nowhere to eat that’s healthy over here, unless you know how to cook… Well, not just around there maybe about five miles away, maybe three or four miles away or more but not right in this area here.” 36-year-old, non-Hispanic Black Female	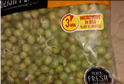
Theme 5. Food Access (Cost) as a HEALth Barrier
“And the easiest stuff to get to at the store’s always the cheaper stuff which is not as healthy so, you kinda wanna, it’s kinda hard ‘cause then you have to take your time to find all the healthy food, weigh the pros and cons in your head.”30-year-old, non-Hispanic White Male	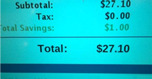
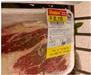	“In the neighborhood per say there’s really nothing other than a Family Dollar and they don’t have fresh fruit and veggies there. We actually have to go somewhere else to get fruit and veggies which at those grocery stores they do have a good variety. Save A lot is down the road as well but it is down the road. In the neighborhood per say there’s really not a lot of options. You have to go outside of your little neighborhood into the bigger area of the community to get the stuff you need.” 45-year-old, non-Hispanic White Female
Theme 6. Time Constraints as a HEALth Barrier
”It makes it more difficult cause you gotta travel further and then you get frustrated with traffic and...Yeah, so.” 30-year-old, non-Hispanic White Male	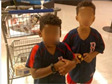
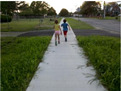	”Obviously if you are out doing something with your kids it is easier to go to a fast food place and getting dinner or if you come home and don’t have a lot of time to make dinner you do the fast options again. Or of you don’t wanna be active you have time to stay home and get stuff ready for dinner to prep to actually cook.” 30-year-old, non-Hispanic White Male
“Well, I work from home! So it’s easier for me, I can take 10–15 minutes and throw something in the crockpot. I don’t have a problem. After they are out of school we can go… I signed them up for t ball soccer, so we are able to do things and be active after school as well.” 49-year-old, non-Hispanic Black Female	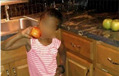
Theme 7. Nutrition Knowledge (or lack of) as a Facilitator/Barrier to HEALth
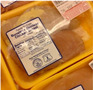	“Some people, some people don’t look at it. I-I-I think a lot of people are are [sic] uneducated on healthy eating….They don’t know how to go in there and look at no meat, or look at this stuff. They’re just buying what they visually see and they think is to buy.”43-year-old, non-Hispanic Black Female
“Salvation Army last year did a cooking class… that engaged the family and the students so everything was hands on…That made me change my ways to look like, hey this is what we can do with this, you can do anything with rice, it all tastes good… You have to get the kids fired up and that makes the parents engaged like OK I see my child very interested in something.” 30-year-old, non-Hispanic Black Female	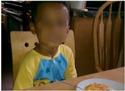
Theme 8. Neighborhood Safety as a Barrier to HEALth
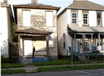	“Um just vacant areas, open areas or open areas that’s just been vandalized. People don’t want to walk to their local grocery stores, or their um neighborhood places because of some of these things.” 43-year-old, non-Hispanic Black Female
Subtheme: Poorly Maintained or Absence of Infrastructure as a Barrier to HEALth
“No, yeah there is a crosswalk, but that’s only there really during, because that’s right where the school is... But it doesn’t cross over to another sidewalk. It crosses over into the grass, so you still have to walk on the edge of the road if you don’t want to walk in the grass.” 30-year-old, non-Hispanic White Male	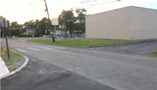
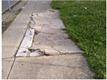	“Where they’re playing… like you said the sidewalks, I wouldn’t let [participant name redacted] ride at a distance by herself because most of the… in our route alone we had to go around trash cans and uneven pavement or absent sidewalks… there’s glass... I hate to see areas that go unattended rather than people with that property or just areas that are just grassy and not maintained.” 45-year-old, non-Hispanic White Female
Subtheme: Crime as a Barrier to HEALth
“They’re actually doing that, I showed you what it looks like now in that little play area. That play area is not on the outside it’s actually on the inside and it’s open. But they have broken ground and they are redoing that whole parking lot area because it had been vandalized and fences had been torn down.” 43-year-old, non-Hispanic Black Female	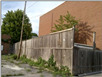
Subtheme: Need for Adult Supervision is a Barrier to HEALth
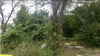	“They’re afraid someone’s going to snatch them up they’ll be in it-you know in the-in the these bushes or in these high weed area. They feel unsafe uh you know allowing them to go there because of some of these concerns.” 43-year-old, non-Hispanic Black Female

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
