# Peer review of "Caregiver Perceptions of Environmental Facilitators and Barriers to Healthy Eating and Active Living during the Summer: Results from the Project SWEAT Sub-Study"

_ijerph, 2021, doi:10.3390/ijerph182111396_

Round 1

Reviewer 1 Report

This outstanding paper examines the sociological, real-life barriers to achieve a healthy lifestyle. The project is well underbuild and explained. I have the following questions:

  • Sub-study: the authors should mention more clearly that recruitement from another trial can yield significant bias (especially when open-label), which is a limiting factor to the overall conclusions.
  • Apart from a low income, could the low level of education also be a confounder? This would mean that people (children, in this case) simply did not get to the necessity of a healthy lifestyle. This is because children approaching their teens buy some of their food themselves these days, being typically prone to industrial machines (soda, candy). The authors should note this in the Discussion.
  • Altogether, as the caregivers were somewhat of sub-investigators, it should be pointed out in the Discussion that conclusions may be skewed (although well-meant).

Reviewer 2 Report

The article deals with an interesting topic, but some aspects should be improved for its publication:

The concept of race is used in an imprecise way. It is not clear whether it refers to physical or biological aspects, and does not state that it is a social construction. Authors should assess the substitution of "racial" minorities for the concept of ethnic minorities. And in the case of not doing so, the use of concepts must be clearly justified.

At a methodological level, a greater analysis of the sample is convenient. It is not specified how many families are from “black” ethnic minorities and how many are “Hispanic”. Likewise, “Hispanic” minorities are presented as homogeneous, regardless of the country of origin of each of these families (it is not the same at a cultural level that a family comes from Puerto Rico, Argentina, Chile, Mexico or Venezuela. Different cultures with different eating habits). This should be extrapolated, if possible, to the analysis of the data and presentation of the results.

The fact that it is students who carry out the fieldwork, rather than an expert researcher, can detract from the rigor of the fieldwork.

The presentation of the results is poor and does not include the variables that have been considered for the selection of informants. It should be specified based on them.

Authorship must be specified in the tables. Table 2 shows some fragments of text enclosed in quotation marks without citing the reference. If they are quotes from informants, it should be clear, also identifying the characteristics of the informant. The source of the photographs must also be cited. In any case, the table format for the content it shows does not seem justified either. The information could better be inserted in the writing.

Round 2

Reviewer 2 Report

Some issues have been clarified by the authors, but there is another that must be resolved:

The clarification that appears in lines 125-128 is insufficient and should be expanded in the wording of the text or in a footnote. It should be explained that the category race is a social construct and not a biological one, as recognized by the American Association of Physical Anthropologists. In reality there is only one race, which is the human race. The categories used, the concepts of race and ethnicity, as well as the imperfections regarding the classifications used should be explained. The article is academic and the definitions must be endorsed by academics, beyond general social definitions. If it is an etic classification, it must be specified. Likewise, the choice of racial minority over ethnic minority remains unclear. In any case, both concepts must be defined.
